# *ERECTA* Modulates Seed Germination and Fruit Development via Auxin Signaling in Tomato

**DOI:** 10.3390/ijms25094754

**Published:** 2024-04-26

**Authors:** Daoyun Chen, Yuqing Xu, Jiawei Li, Hiroshi Shiba, Hiroshi Ezura, Ning Wang

**Affiliations:** 1Graduate School of Life and Environmental Sciences, University of Tsukuba, Tsukuba 305-8572, Ibaraki, Japan; s2130248@u.tsukuba.ac.jp (D.C.); s2330273@u.tsukuba.ac.jp (Y.X.); s2330263@u.tsukuba.ac.jp (J.L.); shiba.hiroshi.gm@u.tsukuba.ac.jp (H.S.); ezura.hiroshi.fa@u.tsukuba.ac.jp (H.E.); 2Tsukuba Plant Innovation Research Center, University of Tsukuba, Tsukuba 305-8572, Ibaraki, Japan

**Keywords:** tomato, receptor-like kinases, auxin, fruit set, seed germination

## Abstract

Tomato (*Solanum lycopersicum*) breeding for improved fruit quality emphasizes selecting for desirable taste and characteristics, as well as enhancing disease resistance and yield. Seed germination is the initial step in the plant life cycle and directly affects crop productivity and yield. ERECTA (ER) is a receptor-like kinase (RLK) family protein known for its involvement in diverse developmental processes. We characterized a Micro-Tom EMS mutant designated as a knock-out mutant of *sler*. Our research reveals that *SlER* plays a central role in controlling critical traits such as inflorescence development, seed number, and seed germination. The elevation in auxin levels and alterations in the expression of *ABSCISIC ACID INSENSITIVE 3* (*ABI3*) and *ABI5* in *sler* seeds compared to the WT indicate that *SlER* modulates seed germination via auxin and abscisic acid (ABA) signaling. Additionally, we detected an increase in auxin content in the *sler* ovary and changes in the expression of auxin synthesis genes *YUCCA flavin monooxygenases 1* (*YUC1*), *YUC4*, *YUC5*, and *YUC6* as well as auxin response genes *AUXIN RESPONSE FACTOR 5* (*ARF5*) and *ARF7*, suggesting that *SlER* regulates fruit development via auxin signaling.

## 1. Introduction

Seed germination, a critical process in plant development, encompasses various physiological and biochemical changes within the seed. The process of seed germination involves testa rupture, endosperm rupture, and radical protrusion. The aleurone layer, the outermost layer of the endosperm, releases enzymes such as amylase, protease, and lipase. These enzymes break down complex molecules like starch, proteins, and lipids into simpler forms, providing essential nutrients for the embryo’s growth [1]. Additionally, seed germination is regulated by a variety of plant hormones. ABA inhibits seed germination by delaying radicle protrusion and endosperm weakening [2]. After the derepression of ABA in endosperm weakening, gibberellin (GA) is a plant hormone necessary for promoting seed germination by facilitating endosperm rupture [3]. Moreover, auxin contributes to the regulation of seed germination by *ARF10/16* to modulate the expression of ABA-sensitive gene *ABI3* indirectly in *Arabidopsis thaliana* [4], thereby influencing the overall germination process. The latest research shows that ARF10/16 also directly binds to the ABA-sensitive transcription factor ABI5 to activate the transcriptional function, finally inhibiting seed germination in *Arabidopsis thaliana* [5]. Auxin in seeds mainly originates from the early developmental stage of endosperm. *YUCs* are primarily induced in the endosperm immediately post-fertilization to synthesize auxin [6]. More than half of these auxins synthesized during endosperm development are retained in mature seeds [7].

Auxin serves as a master regulator of plant growth and development, orchestrating a wide range of cellular processes to ensure proper plant growth, morphology, and adaptation to environmental conditions. Its significance lies in its ability to regulate numerous physiological processes, including root development, apical dominance, cell elongation, and tissue differentiation. In land plants, auxin synthesis mainly relies on the tryptophan aminotransferase (TAA)/YUC pathway using tryptophan as the precursor [8]. *YUCs* in tomato have not been systematically cloned; a phylogenetic analysis shows there are at least nine *SlYUC* genes in the tomato genome; among them, *SlYUC1*, *SlYUC3*, *SlYUC4*, *SlYUC5*, and *SlYUC6* are expressed in the shoot [9]. During flowering, the auxin biosynthesis genes *FZY2 *(*YUC4* in this study, Appendix A), *FZY3* (*YUC7*), and *FZY6* (*YUC6*) have relatively higher expression levels in floral organs [10]. Auxin signaling depends on the transcription factor *Auxin Response Factor* (*ARF*) family. The F-box protein TRANSPORT INHIBITOR RESISTANT (TIR) that binds to auxin promotes the ubiquitination of AUXIN/INDOLE ACETIC ACID (Aux/IAA), thereby derepressing the inhibition of ARF by IAA and promoting the downstream expression of auxin-sensitive genes [11]. The genome-wide analysis finds at least 17 *SlARF* genes from the tomato genome [12].

In tomato breeding, the focus on improving fruit quality revolves around selecting tomatoes with desirable characteristics and taste. Additionally, enhancing resistance to cracking or diseases, along with overall yield, are important considerations. Understanding and optimizing genetic factors are essential for maximizing fruit sets and ensuring high yields in tomato production. The process of natural fruit set in tomatoes is initiated by pollination, during which the transfer of pollen grains stimulates the ovary to synthesize substantial quantities of auxin [13]. Furthermore, a localized increase in auxin concentration within the ovary of tomato plants leads to the development of seedless and enlarged fruits [14]. In addition, silencing the auxin-responsive gene *IAA9* can produce seedless and larger-sized tomato fruits [15]. Recent research has confirmed that ARF7 directly binds to IAA9. Experiments involving the use of CRISPR-Cas9 to knock out *ARF7* have resulted in the production of seedless, enlarged fruits that are similar to the mutant *iaa9*, also resulted in increased expression of *ARF2B* and decreased expression of *ARF5* and *ARF8B* [16]. The mutants of *ARF5* can cause seedless and reduce fruit size [17]. The ARF family coordinately regulates fruit development; double mutations in *ARF5* and *ARF7* cause parthenocarpy [16,17]. Excessive seed content in tomatoes can negatively impact the taste [18]. Beyond seedlessness, another taste-related concern arises from seed-sprouting within the fruit. Although injecting ABA into the fruit can inhibit seed-sprouting [19], it also promotes ethylene synthesis, hastening fruit ripening [20]. Addressing seed-sprouting without accelerating ripening presents a critical challenge for researchers.

Ethyl methanesulfonate (EMS) alkylates the guanine base as a commonly used chemical mutagen, leading to primarily C/G to T/A base transitions [21]. EMS acts by causing point mutations in the DNA sequence by alkylating guanine residues, which may result in altered phenotypes such as changes in plant morphology, physiology, or biochemical pathways. EMS mutagenesis is a powerful tool in plant breeding for generating genetic diversity and creating novel traits that can contribute to the development of improved crop varieties with enhanced agronomic traits. The mutagenized populations are typically subjected to phenotypic screening to identify individuals with the desired traits, bypassing the controversy of transgenic [22]. As a model plant of the *Solanaceae* family, Micro-Tom has also been utilized to generate and establish a substantial EMS mutant population. The Tomato Mutant Archive TOMATOMA (University of Tsukuba) hosts approximately 13,000 tomato EMS mutant lines, which were established via the National BioResource Project (NBRP) [23]. This vast repository of mutant materials offers valuable resources for tomato breeding, screening, and functional genome research.

*ER* encodes RLK family protein characterized by its extracellular leucine-rich repeats (LRR) domain, transmembrane domain, and intracellular kinase domain, which is a pleiotropic gene that has been studied in *Arabidopsis thaliana*. *ER* plays a crucial role in various developmental processes; *AtER* along with its homologous genes *AtERL1* and *AtERL2* redundantly regulates stomatal genesis [24], cell division, and the differentiation of apical meristems [25], ovule development [26], and regulation of seed size [27]. In tomato, the *SlER* family is only composed of two genes: *SlER* and *SlERL1* [28]. The miniaturization of crops has always been one of the main breeding goals. Compact crops, with dense foliage and inflorescences, can be planted more densely to increase field utilization. Genetic studies have explored the use of CRISPR/Cas9 technology to suppress *SlER* expression in the M82 background, which has produced compact tomatoes with more condensed inflorescences suitable for denser planting [29]. The transgenic tomato plants with overexpressed *ER* resulted in significantly increased heat tolerance, attributed to the abundance of ER protein, which influenced the ratio of the number of stomata to the total number of leaf epidermal cells [30]. In the process of seed germination, *ER* has been confirmed to play a pivotal role in regulating seed germination under salt stress conditions via ABA-sensitive genes *ABI3* and *ABI5* signaling pathways [31]. In tomato, the functions of *ER* on seed germination and fruit development are still poorly studied. Furthermore, *ER* impacts various aspects of auxin dynamics, including its biosynthesis [32], transportation [33], and downstream signaling cascades [34].

In this study, we characterized a Micro-Tom EMS mutant designated as *sler*, previously characterized as a knock-out mutant in a prior investigation [29]. Upon examination, we observed phenotypic traits reminiscent of those observed in the *sler* mutant, including dwarfism, compactness, and reduced seed size. Our research endeavors were primarily directed towards unraveling the involvement of auxin intervention in tomato fruit development and seed germination processes. For the first time, this study explains the mechanism by which *ER* inhibits seed germination by regulating auxin synthesis. Notably, our findings not only shed new light on the role of *ER* in the auxin synthesis pathway but also prompt a re-evaluation of the potential contributions of the *ER* gene to tomato breeding efforts.

## 2. Results

### 2.1. sler Exhibited Compact Traits

Plant height, internodal length, stem diameter, and pedicel length were measured to determine the extent of *sler* dwarfism (Figure 1) at 0 days after flowering (DAF0). The average plant height of the *sler* was reduced to 55.71 ± 4.02 mm, a significant decrease compared to WT’s height of 89.94 ± 9.60 mm (Figure 1E). On average, the internode length of *sler* at DAF0 was 5.53 ± 0.40 mm, which is shortened significantly compared with the internode length of WT 14.48 ± 1.49 mm (Figure 1F). The stem diameter of the *sler* at DAF0 was 7.81 ± 0.49 mm on average and thickened significantly compared with the stem diameter of WT 5.29 ± 0.75 mm (Figure 1G). The pedicel lengths of WT and *sler* were 4.93 ± 0.66 mm and 0.89 ± 0.05 mm, respectively. The pedicel length of the *sler* was significantly shorter than WT (Figure 1H).

### 2.2. sler Altered Fruit and Seed Formation

The fruit weight of the *sler* exhibited a significant reduction compared to that of the WT (Figure 2A,B). The average heights of WT fruits were recorded at 20.64 ± 1.85 mm, whereas those of *sler* fruits averaged 15.83 ± 1.88 mm (Appendix A). The heights of *sler* fruits were significantly lower than WT. The average widths of WT fruits were recorded at 21.13 ± 1.79 mm, whereas those of *sler* fruits averaged 19.01 ± 2.77 mm (Appendix A). The average weights of WT fruits were recorded at 5.24 ± 1.19 g, whereas those of *sler* fruits averaged 3.5 ± 1.41 g (Appendix A). The number of seeds in *sler* mutant fruit was notably lower than that of WT. On average, WT fruits contained 34.67 ± 5.80 seeds per fruit, whereas *sler* mutant fruits contained 7.22 ± 4.86 seeds per fruit (Figure 2H). The germination rates showed significant differences between the WT and *sler* mutants, which were 79.6% and 35.6%, respectively (Figure 2G). In soil, the germination rate of 38.8% for *sler* was also lower than 75.5% for WT (Appendix A). When the pollen tube elongates, it produces a large amount of callose to ensure that its cell wall can resist the circumferential tension stress to reach the ovule [35]. Callose can be visualized by aniline blue staining [36]. Our microscopic examination observed elongated pollen tubes and abundant callose fluorescence signals revealed that pollen tubes of *sler* successfully germinated and elongated (Figure 2D). Distinct elongated ovules, differing from those observed in the WT, were discernible within the ovary of the *sler*, as illustrated in Figure 2E,F. The seeds of the WT exhibit noticeably fuller grains compared to those of the mutant (Appendix A). Detailed measurements revealed significant differences in seed length and width between WT and the *sler* mutant. Specifically, the seed length of WT averaged at 3.90 ± 0.07 mm, whereas that of the *sler* mutant was significantly shorter at 3.50 ± 0.07 mm. Similarly, the seed widths of WT and *sler* mutant exhibited substantial discrepancies, with WT seeds measuring 2.57 ± 0.08 mm in width, while *sler* mutant seeds were notably narrower at 1.93 ± 0.04 mm.

### 2.3. Genetic Redundancy between SlER and AtER

According to our sequencing results of *sler*, we identified a mutation in the 3164th adenine of *ER*, where it was replaced by thymine. This resulted in converting the 398th lysine into a stop codon, leading to a knockout mutation of *ER* (Figure 3A). We conducted phylogenetic analysis to gain insight into the evolutionary relationships among species via genetic changes (Figure 3B). We compared the amino acid sequences of *SlER* and its homologous gene *SlER-like1* (*SlERL1*, *Solyc03g007050*) with their orthologous genes in *Arabidopsis thaliana*. The results revealed that *SlER* is genetically closest to *AtER* (*AT2G26330*), whereas *SlERL1* is more closely related to *AtER-like1* (*AtERL1*, *AT5G662230*) and *AtER-like2* (*AtERL2*, *AT5G07180*). Based on the expression data from the Tomato eFP Browser (https://bar.utoronto.ca/efp_tomato/cgi-bin/efpWeb.cgi (accessed on 10 July 2023)), it is evident that the expression level of *SlER* is significantly higher than that of *SlERL1* across all tissues of tomato (Figure 3C). Our qPCR analysis revealed the expression of *SlER* in tissues such as the pedicel, top apical, and stem, which are not covered by the existing database (Figure 3D). Furthermore, the decreased relative expression in roots observed in our study aligns with the expression trend observed in the database. Additionally, our result revealed that the expression of *SlERL1* in the *sler* mutant did not show a significant increase compared to the WT (Figure 3E).

### 2.4. Enhanced Auxin Signal in the Seed and Ovary

Due to the lower germination rate observed in *sler* seeds, we conducted an examination of hormonal signals within the seeds. To investigate auxin distribution during seed germination, we performed immunolocalization studies of longitudinal sections with an anti-indole-3-acetic acid (IAA) monoclonal antibody and observed a higher immunofluorescence signal in the area of the *sler* endosperm when compared to WT (Figure 4A). Considering that auxin synthesis in seeds is fertilization-induced and occurs in the endosperm immediately post-fertilization [6]. Therefore, for the auxin retained in the seed, its synthesis and gene regulation must be detected during the ovary stage; we further investigated the presence of auxin in the ovary. Our analysis revealed that the auxin fluorescence signal in the ovary of *sler* was higher compared to that of WT (Figure 4B).

We employed qPCR analysis to assess the expression levels for critical genes involved in fruit development and auxin synthesis (Figure 5A). In the ovary, the expression level of *ARF5* in the *sler* was significantly lower than that of the WT, while the expression level of *ARF7* was significantly higher. In addition, the expression of *IAA9* was not significantly different between WT and *sler* (Appendix A). Additionally, the expression levels of *YUC1*, *YUC4*, *YUC5*, and *YUC6* of *sler* were significantly elevated relative to those in the WT. Among them, the expression of *YUC1* and *YUC4* in *sler* upregulated significantly, reaching 9.73 times and 3.07 times those of WT. Furthermore, we observed the expressions of *ABI3* and *ABI5* in WT and *sler* seeds. The expression level of *ABI3* and *ABI5* in WT seeds was significantly downregulated from 24 h of soaking in water and dropped to an extremely low level after 48 h of soaking in water. Correspondingly, in *sler*, the expression level of *ABI3* and *ABI5* was not significantly downregulated until 72 h after soaking in water, and the expression level of *ABI3* was even upregulated after 48 h of soaking in water (Figure 5B).

## 3. Discussion

### 3.1. Gene Duplication and Functional Divergence of ERECTA

The evolution of a genome is an intricate process that involves various genetic alterations such as mutations, duplications, rearrangements, and horizontal gene transfers. These changes play a significant role in shaping the genetic makeup of organisms over time. As a result, homologous genes may undergo functional diversification, leading to the creation of new genes with novel functions (neofunctionalization), or their function may be divided (subfunctionalization) [37]. Previous studies demonstrated that *AtER*, along with its homologous genes *AtERL1* and *AtERL2*, redundantly control various traits in *Arabidopsis thaliana*. Loss of function mutations in *AtER* affects inflorescence length, which can be partially rescued by *AtERL1* or *AtERL2* expression. However, mutations in *AtERL1* and *AtERL2* exacerbate the compact phenotype [38]. During ovule development, only one megaspore mother cell is produced in an ovule, which divides into four haploid megaspores. At least one of the spores develops into haploid female gametophytes [39]. Single mutations in *AtER* do not affect seed number; triple mutants (*ater*, *aterl1*, and *aterl2*) lead to a reduction in seed number due to multiple megaspore mother cells being differentiated in one ovule [26,40]. Under salt stress, double (*ater* and *aterl1* or *aterl2*) and triple mutants exhibit delayed seed germination, whereas single *ater* mutants show no significant changes [31]. Therefore, it appears that the functional diversification of the *ER* family is attributed to subfunctionalization in *Arabidopsis thaliana*.

Nevertheless, the process of functional diversification of duplicated *ER* genes differs in tomato. The evolutionary fate of duplicated genes involves either acquiring new functions or becoming nonfunctional. In most cases, the duplicated genes are free to acquire degenerative mutations and become pseudogenes (pseudogenization) because there are no functional constraints, and the genes are not under selection pressure. Our investigations have revealed that the mutation of *sler* alone has led to abnormal development of ovules, reduction in seed number, and decreased seed germination rate (Figure 1 and Figure 2). These phenotypic characteristics are not observed in *ater* single mutant and can only be observed when at least two genes in the *AtER* family are mutated together. In functional terms, kinases often exhibit dosage-dependent behavior, wherein those with elevated expression levels exert a dominant influence [41]. Taking into consideration the significantly lower expression level of *SlERL1* compared to *SlER* (Figure 3C), along with the observation that its expression does not increase in *sler* (Figure 3E), we infer that the function of *ER* in tomatoes is not as redundant as the orthologous genes in *Arabidopsis thaliana*. Throughout the course of tomato genome evolution, *ER* has acquired crucial functionalities inherited from ancestral genes. Conversely, the *ER-like* gene in tomato has undergone functional degradation, rendering it relatively lower gene expression levels and minor functionality.

### 3.2. ERECTA Regulates Seed Germination via Auxin Signaling

Seed germination is a critical physiological process in spermatophytes and is inhibited by the phytohormone ABA [42], and auxin has been shown to enhance the inhibition of germination by ABA via the ABA-sensitive genes *ABI3* and *ABI5* [5,43]. In this study, we observed a reduction in the germination rate of *sler* mutant seeds (Figure 2G). The seed size of the *sler* is smaller than that of WT (Appendix A), which may cause the endosperm to shrink and fail to provide sufficient energy for hypocotyl elongation. Additionally, a plausible explanation could be the accumulation of a substantial amount of auxin within the seeds (Figure 4A). *AtER* has been identified as a key regulator of seed germination under salt stress via *ABI3* and *ABI5* ABA signaling [31]. Additionally, the expression level of *ABI3* and *ABI5* in WT decreased in seeds soaked in water from 24 to 72 h. However, the expression level of *ABI5* in *sler* mutant seeds remained unchanged. Notably, the expression of *ABI3* was upregulated in 48 h, indicating that excess auxin in *sler* endosperm upregulated the expression of *ABI3* to inhibit seed germination (Figure 5B). These results suggest that the accumulation of auxin in the seed endosperm is regulated by *SlER*, which in turn regulates the expression of ABA-responsive genes *ABI3* and *ABI5*, thus controlling seed germination. Additionally, auxin synthesis in seeds initiates during the early stages of endosperm cell development post-fertilization [7,44]. Our finding revealed higher levels of auxin content in the ovary and ovules of the *sler* mutant compared to the wild type (Figure 4B), along with the expression upregulation of *YUC1*, *YUC4*, *YUC5*, and *YUC6* in the *sler* ovary (Figure 5A), suggesting that *SlER* play a role in regulating auxin content in seeds starting from the early stages of fruit development. Previous studies have shown that *YUC7* (corresponds to gene *FZY3*, Appendix A) is relatively higher expressed during the flowering stages [10]. However, the expression of YUC7 in the ovary was not detected in our study (Figure 5A). The results suggested that *YUC7* exhibits spatial expression specificity. In *Arabidopsis thaliana*, the *ater aterl1 aterl2* triple mutant significantly decreases germination rate under salt stress conditions [31]. Still, in this study, the single gene *SlER* knock-out leads to a low germination rate without salt stress (Appendix A). In a pan-species seed conservation study, it was found that short-lived species are more likely to break seed dormancy [45]. Considering that *Arabidopsis thaliana* completes its life cycle in 8–10 weeks from germination to harvesting [46], we speculate that the extremely short life cycle of *Arabidopsis thaliana* results in the inhibition of seed germination by *er* mutations not being as obvious as in tomato. Studies have shown that *AtER* regulates hypocotyl elongation after germination [32]. Re-examining the effect of *SlER* on auxin signaling in seedlings after germination in tomato will be a future task.

The impact of *ER* on the auxin pathway has remained ambiguous. Previous studies have shown that the *DR5::GFP* fluorescence signal is reduced in the *ater* mutant [32]. This contradicts the increased auxin accumulation observed in the *sler* mutant in our study, where we demonstrated elevated auxin levels in both the ovary and seeds (Figure 4A,B). Given that the *DR5* is the core TGTCTC hexamer sequence in the *ARF* family DNA-binding promoter it exerts its function via binding with ARF transcription factors [11], and the transcription factor *BRASSINOZOLE RESISTANT 1* (*BZR1*), whose expression is responsive to brassinosteroids (BR) and regulated by *ER*, can bind to certain *ARF* family gene promoters to upregulate *ARF* expression [26]. In tomato, the expression of *ARF5* is regulated by *BES1-interacting MYC-like protein 1* (*BIM1*), a co-regulator of *BZR1* [47], which is also consistent with the downregulation of *ARF5* expression in our qPCR results (Figure 5A). We speculate that the observed decrease in the *DR5:GFP* signal in the *ater* mutant from a previous study was likely due to a decrease in *ARF* expression rather than a decrease in auxin biosynthesis. Our investigations provide a new perspective on elucidating the involvement of *ER* in the auxin pathway.

### 3.3. ERECTA Regulates Fruit Set

Tomato fruit development is directly linked to auxin signaling. *IAA9* is a regulator of fruit development, and its functional inhibition can lead to parthenocarpy [15]. In this study, the expression of *IAA9* did not change (Appendix A), indicating that *IAA9* is not involved in the pathway regulated by *SlER*. *ARF5* and *ARF7* cooperate to regulate parthenocarpy in tomato. *ARF7*, primarily expressed during the flowering stage, acts as a repressor in fruit development. The suppression of *ARF7* expression leads to the enlargement of fruit and the absence of seeds [16]. However, unlike *ARF7*, inhibition of *ARF5* expression will lead to smaller fruits [17]. In this study, we observed a downregulation of *ARF5* expression and an upregulation of *ARF7* expression in the *sler* ovary (Figure 5A). These changes in *ARF* expression are associated with smaller fruit size. In fact, the fruits of the *sler* in this study were indeed smaller than those of the WT (Appendix A). This suggests that *SlER* regulates tomato fruit development via the auxin signaling pathway. With respect to seed number, the increase in ovary auxin and the decrease in *ARF5* expression in *sler* are expected to result in a decrease in seed number, while the increase in *ARF7* expression is anticipated to lead to an increase in seed number [14,16]. Under the combined regulation of these three factors, the seed number of *sler* was significantly lower than that of WT, indicating that *ARF5* has a superior function in regulating seed number than *ARF7*. This study implies that *ER* regulates tomato fruit development via auxin signaling in a multi-factorial manner.

The fruits of the *Brassicaceae* family are siliques, which naturally split open to release seeds, rendering them less suitable for consumption. In contrast, tomato fruits, which are classified as berries and constitute the primary edible parts, make the *sler* an excellent model for studying the impact of *ER* on fruit development. In the M82 background, previous research has utilized CRISPR/Cas9 to knock out *SlER*, resulting in tomato varieties that can be planted more densely [29]. In this study, we discovered that knockout mutant *sler* is characterized by reduced fruit size and seed number, which could impact fruit quality and complicate seed preservation. Given that CRISPR/Cas9 technology tends to produce knockout mutations, screening for less severe *SlER* missense mutations via EMS mutagenesis might be more advantageous for breeding.

In summary, our study reveals that *SlER* singularly governs phenotypes such as inflorescence development, seed number, and seed germination, which are redundantly regulated by *AtER*, *AtERL1*, and *AtERL2* in *Arabidopsis thaliana*. This suggests that while *ER* is functionally conserved within the tomato genome, it has not undergone subfunctionalization. We observed an elevation in auxin levels and alterations in the expression of *ABI3* and *ABI5* in *sler* seeds compared to the WT, indicating that *SlER* modulates seed germination via auxin and ABA signaling. Additionally, we detected an increase in auxin content in the *sler* ovary and changes in the expression of auxin synthesis genes (*YUC1*, *YUC4*, *YUC5*, and *YUC6*) as well as auxin response genes (*ARF5* and *ARF7*), suggesting that *SlER* regulates fruit development via auxin signaling. Our findings suggest *SlER* orchestrates seed germination and fruit development via modulation of auxin and ABA signaling pathways and advocate for the screening of *SlER* knockdown mutants as potential breeding materials.

## 4. Materials and Methods

### 4.1. Plant Materials and Growth Conditions

The Micro-Tom wild type and *sler* mutant were obtained from the National Bioresource Project tomato (NBRP tomato: http://tomato.nbrp.jp/indexEn.html (accessed on 6 May 2017)). The *sler* mutant strain from the Tomato Mutants Archive (TOMATOMA: https://tomatoma.nbrp.jp/ (accessed on 6 May 2017)) has the identification number TOMJPE5066-1. The plants were grown in a glasshouse located at the Gene Research Center (GRC) of Tsukuba-Plant Innovation Research Center (T-PIRC), University of Tsukuba, from March 2023 to June 2023 with controlled environmental conditions of 24 °C temperature, 42% humidity, 16 h of light (12300LUX), and 8 h of darkness, fertilized once a week with 500-fold diluted Hyponex Undiluted Solution (HYPONeX, Osaka, Japan).

### 4.2. Phenotypic Measurement

Plant height, stem diameter, internode length, and pedicel length were investigated in DAF0 (the day after flowering). The height of the plant was measured based on the distance from the ground to the first flower cluster. The internode length was measured by DAF0, the distance from the first lateral branch on the ground to the first lateral branch in the first inflorescence, and the length/(number of lateral branches − 1) was used for statistics. The diameter of the stem was measured just below the first inflorescence with DAF0. Pedicel length is measured at DAF0 from the sepal base to the joint. Pollination is carried out at DAF0 via an electric vibrator TS-550 (TAKII, Tokyo, Japan). Then, we visually confirmed enough pollen on the stigma to ensure the number of seeds per fruit and fruit weight were measured under sufficient pollination. Fruit height was measured from the stigma to the top of the tomato, where it was linked to the pedicel. Fruit width was measured by measuring the maximum distance from one side of the tomato fruit to the other on a horizontal plane. Seed size measurements were performed with ten seeds as one biological replicate to reduce measurement errors.

### 4.3. Seed Germination Test

Seeds of WT and *sler* were placed in a Petri dish containing a piece of water-soaked filter paper, 15 seeds in one Petri dish considered as a replicate, three replicates each of WT and *sler*. All Petri dishes were placed in a dark room at 25 °C and 42% humidity, and the germination rate was counted after seven days.

We also conducted a germination test in soil. A total of 49 seeds, each of WT and *sler*, were sown into water-absorbing jiffy mix soil (Seedfun, Yokohama, Japan), placed in the same glasshouse located at the GRC, and the number of germinated seeds was counted 10 days later.

### 4.4. Aniline Blue Staining of Pollen Tubes

The pistil styles were immersed in a decolorizing/fixing solution (acetic acid:ethanol = 1:3) 24 h after pollination and allowed to decolorize and fix for one hour. Subsequently, the fixed pistil style was immersed in 1N NaOH solution and incubated at 60 °C for 2 h to facilitate tissue softening. Following the removal of the 1N NaOH solution, a 0.01% aniline blue solution (prepared by dissolving 0.01% aniline blue in 2% K_3_PO_4_ solution, left overnight, and stored in an amber bottle) was applied and allowed to stain at room temperature for 2 h.

### 4.5. DNA Extraction and Sequencing

Tomato leaves were mixed with 150 μL of crushing buffer (1M NaCl, 100mM Tris-Cl pH 8.0, 50mM EDTA) and homogenized using a Tissue Lyser II (QIAGEN, Hilden, Germany). An amount of 150 µL of 2× CTAB solution (6% CTAB, 1M NaCl, 100mM Tris-Cl pH 8.0, 50mM EDTA) and one µg of RNase (NIPPON GENE, Tokyo, Japan) were added and heated at 65 °C for 60 min. After cooling to room temperature, 150 μL of chloroform was added and centrifuged at 12,000 rpm for 10 min. An amount of 200 μL of the upper layer was transferred to a new microtube, 200 μL of isopropanol was added, and mixed. The mixture was centrifuged at 12,000 rpm for 10 min, the supernatant was discarded, 500 μL 70% ethanol was added, and the mixture was centrifuged at 12,000 rpm for 5 min. The supernatant was discarded and dried for 10 min. Then, 100 μL of pure water was added and dissolved.

We amplified the sequence of *SlER* using gene-specific primers (Appendix A). PCR products were purified using a QIAquick PCR Purification Kit (QIAGEN, Hilden, Germany) according to the manufacturer’s instructions. The PCR product was used as the template for sequencing with the BigDyeKit (Applied Biosystems, Waltham, MA, USA).

### 4.6. Comparative Analysis of ER Family Genes

The protein sequences of *Arabidopsis thaliana* and tomato were obtained from the National Center for Biotechnology Information (NCBI: https://www.ncbi.nlm.nih.gov/ (accessed on 15 July 2023)) and Sol Genomics Network (https://www.solgenomics.net/ (accessed on 15 July 2023)). Subsequently, an un-rooted Neighbor-Joining phylogenetic tree was constructed by aligning all the ER family protein sequences of tomato and *Arabidopsis thaliana* with the MEGA11 11.0.13 program.

Furthermore, the expression level data of *SlER* and *SLERL1* were obtained from the Tomato eFP Browser (https://bar.utoronto.ca/efp_tomato/cgi-bin/efpWeb.cgi (accessed on 10 July 2023)). These data were then utilized to generate graphical representations for visual analysis.

### 4.7. Immunolocalization of IAA

The seed and ovary samples were soaked in 4% (*w*/*v*) paraformaldehyde overnight at 4 °C. Then, samples were dehydrated using a gradient series of sucrose solutions (15%, 30% [*v*/*v*]). After dehydration, samples were embedded with Tissue-Tek O.C.T. Compound (SAKURA, Tokyo, Japan) and sliced into 40 µm slices by the CM1950 cryostat (Leica). The slide was observed by AXIO Imager.A2 (ZEISS, Baden-Württemberg, Germany) and photographed by AxioCam ERc5s (ZEISS). The sections were incubated with 1:500 (*v*/*v*) dilutions of antiIAA monoclonal antibody (Sigma-Aldrich, St. Louis, MO, USA, A0855) overnight at 4 °C, and then with DyLightTM 488-labeled anti-mouse IgG antibody (1:500 [*v*/*v*], KPL, 5230-0391) for at least 4 h at room temperature in the dark. The fluorescence signal was recorded using a laser scanning confocal microscope LSM700 (Leica, Wetzlar, Germany), with an excitation wavelength of 488 nm and emission wavelength of 518 nm [48].

### 4.8. RNA Extraction and qRT-PCR

Ovary at the DAF0 stage and seeds soaked in water for 12, 24, 48, and 72 h were immediately frozen in liquid nitrogen and stored at 80 °C until use. Total RNA was extracted using the TRIzol™ Reagent (Invitrogen, Waltham, MA, USA) following the manufacturer’s protocol. An amount of 0.5 micrograms of total RNA was reverse transcribed into cDNA using the ReverTra ACE™ qPCR RT Master Mix with gDNA Remover (TOYOBO, Osaka, Japan). The RT-qPCR was performed using a StepOne Plus real-time PCR system (Applied Biosystems). Transcript quantification was dependent on the 2^−ΔΔCt^ analysis method [49]. *Sl-Ubiquitin* was used as the internal reference gene.

### 4.9. Statistical Method

All statistical processing in this article was performed using OriginPro (10.1.0.170) software. Multiple comparison was performed via Tukey’s test in Figure 5B. Significant differences in Figure 1E–H, Figure 2G,H, Figure 3D,E, and Figure 5A were tested using a two-tailed Student’s *t* test.

## Figures and Tables

**Figure 1 ijms-25-04754-f001:**
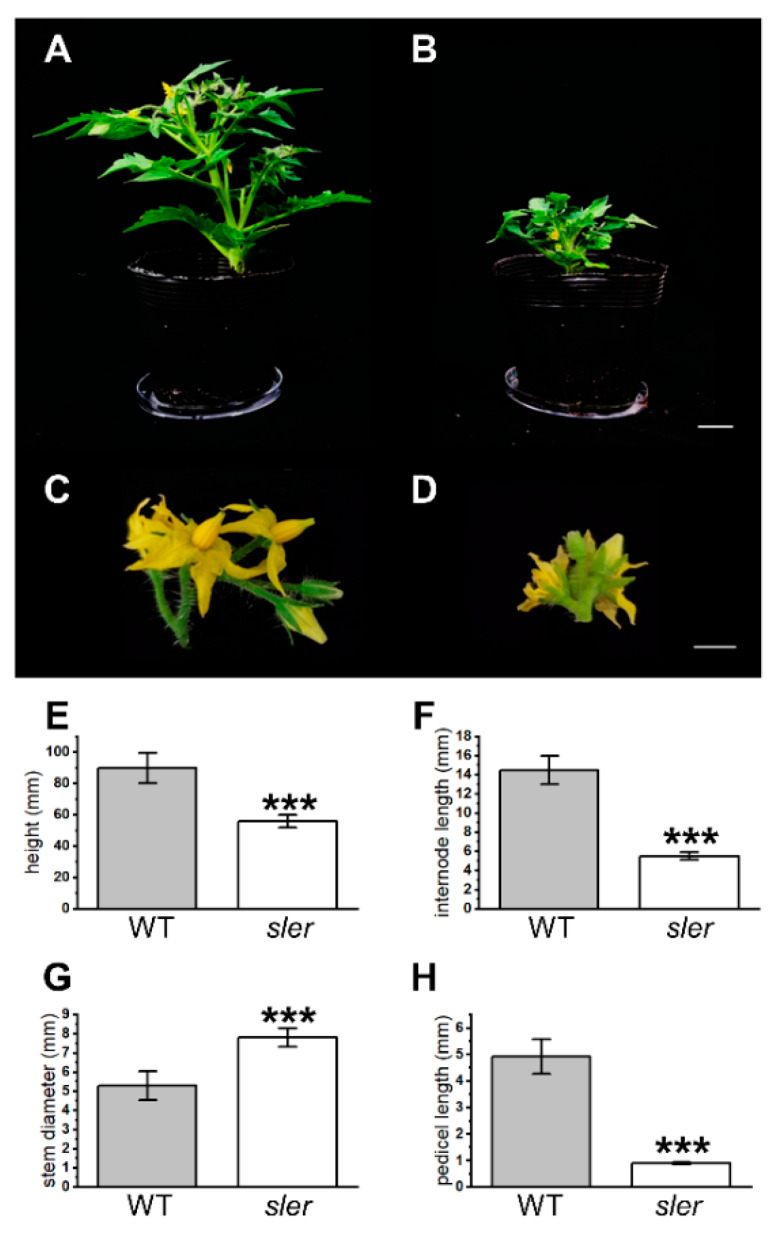
Dwarf trait of *sler.* Plants of WT (**A**) and *sler* (**B**), bar = 2 cm. Inflorescence of WT (**C**) and *sler* (**D**), bar = 5 mm. Height of WT and *sler* (**E**). Internode length of WT and *sler* (**F**). Stem diameter of WT and *sler* (**G**). Pedicel length of WT and *sler* (**H**). Error bars represent the standard deviations. Significant differences were determined by two-tailed Student’s *t* test (***, *p* < 0.001), *n* = 8.

**Figure 2 ijms-25-04754-f002:**
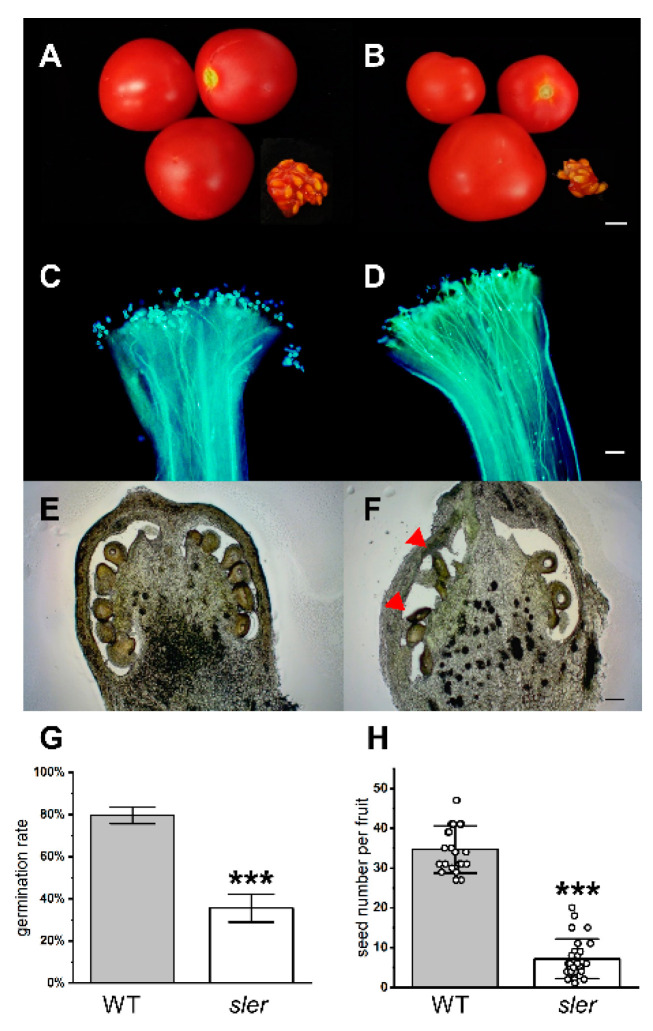
Fruit and seed traits of *sler.* Fruit size and seed number of WT (**A**) and *sler* (**B**), bar = 5 mm. Pollen tube elongation of WT (**C**) and *sler* (**D**), bar = 100 μm. Ovary sections at DAF0 of WT (**E**) and *sler* (**F**); Arrows indicate the slender ovule, bar = 100 μm. The percentage of seeds germinated in the Petri dish (**G**), *n* = 3. Seed number per fruit of WT and *sler* (**H**), *n* = 24. Error bars represent the standard deviations. Significant differences were determined by a two-tailed Student’s *t* test (***, *p* < 0.001).

**Figure 3 ijms-25-04754-f003:**
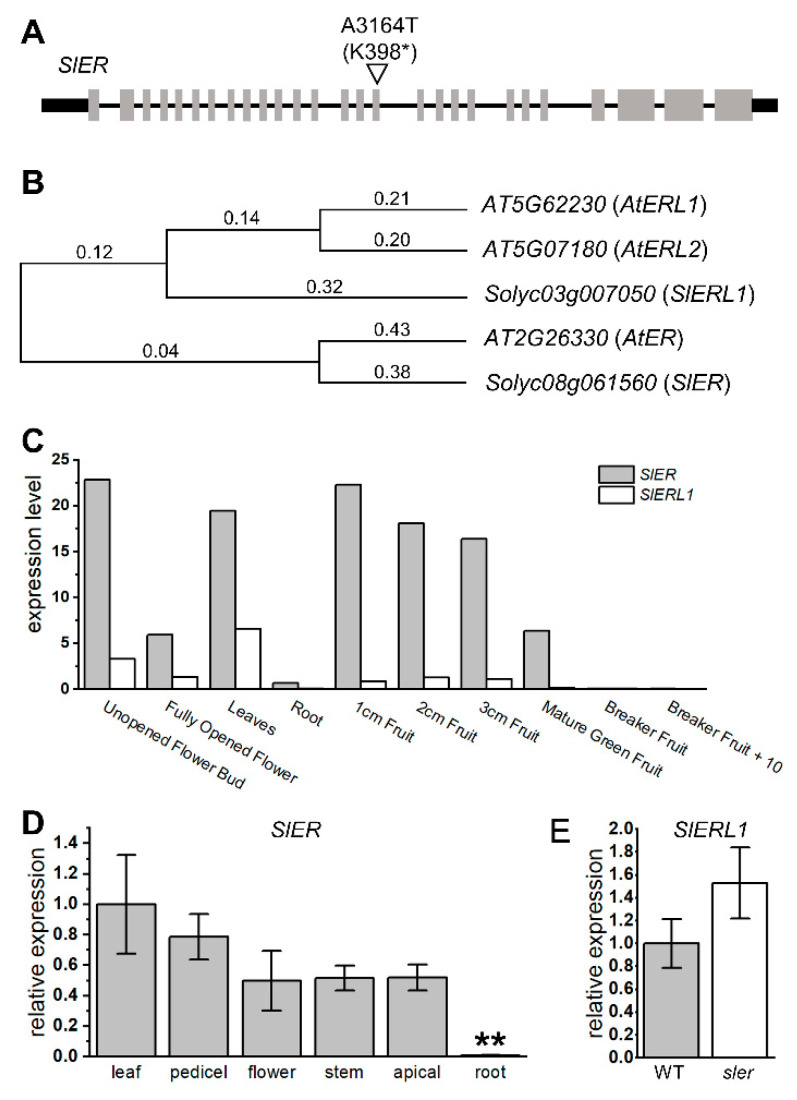
Mutation sites of *sler* and expression levels of *SlER* family. Mutation allele in *SlER* of *sler* (**A**). The “*” represents stop codon. Phylogenetic tree of *ER* (**B**). Expression level of *SlER* and *SlERL1* from Tomato eFP Browser (**C**). Relative expression of *SlER* in leaf, pedicel, flower, stem, apical, and root (**D**). Significant differences were determined by two-tailed Student’s *t* test (**, *p* < 0.01), *n* = 4. Relative expression of *SlERL1* between WT and *sler* pedicel (**E**). *n* = 4. Error bars represent the standard deviations.

**Figure 4 ijms-25-04754-f004:**
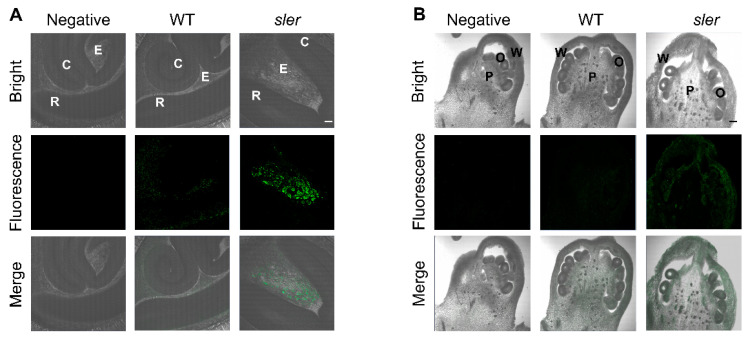
Auxin level in seed and ovary. Immunolocalization of IAA in the seed after 12 h soaked in water (**A**), and Immunolocalization of IAA in the ovary of flowering floral structure (**B**). Immunofluorescence assays were performed by probing with an anti-IAA monoclonal antibody as primary antibody and fluorescent labeling with DyLightTM 488-labeled anti-mouse IgG antibody as secondary antibody. Fluorescently labeled tissues were observed under an excitation wavelength of 488 nm and an emission wavelength of 518 nm. The negative control utilized a secondary antibody (DyLightTM 488-labeled anti-mouse IgG) instead of the anti-IAA monoclonal antibody. C, cotyledon; E, endosperm; R, radical. P, pistil; O, ovule; W, ovary wall. Bar = 100 μm.

**Figure 5 ijms-25-04754-f005:**
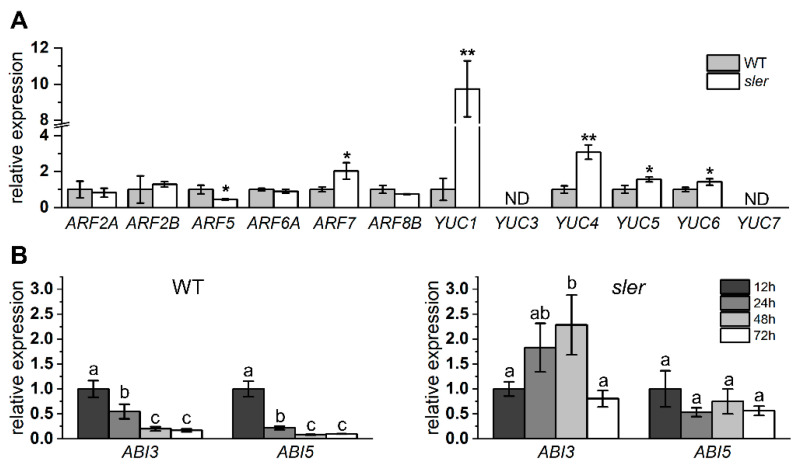
Relative expression of auxin and ABA pathway genes. Expression of *ARF* and *YUC* genes in the ovary at DAF0 (**A**). The “ND” represents expression level was not detected. Significant differences between WT and mutants were determined by two-tailed Student’s *t* test (* 0.01 < *p* < 0.05; ** 0.001 < *p* < 0.01), *n* = 4. The expression levels of the *ABI3* and *ABI5* genes in the seeds were assessed at different time points (12, 24, 48, and 72 h) following immersion in water (**B**). Error bars represent the standard deviations. Letters indicate significant differences according to Tukey’s test (*p* < 0.05), *n* = 3.

## Data Availability

The authors declare that all data supporting the findings of this study are available within the article and its Appendix A.

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
