# Peer review of "ERECTA Modulates Seed Germination and Fruit Development via Auxin Signaling in Tomato"

_ijms, 2024, doi:10.3390/ijms25094754_

Round 1
Reviewer 1 Report
Comments and Suggestions for Authors
In this manuscript, the authors reported evidence of ER gene in tomato regulating seed germination via auxin and abscisic acid signaling and fruit development via auxin signaling. Scope of the study is acceptable where the authors have used results from different analyses, e.g., gene expression analysis, microscopical observation and phenotypic trait examination to support their conclusions. In general, the manuscript is well-prepared, and the authors have carefully reported details of the experiments they performed. The results obtained were generally well-discussed. Gap of knowledge to be filled by the study and novelty of the findings were clearly highlighted in the last two paragraphs of INTRODUCTION. I have only very minor feedback as listed below.
Below is my feedback for the authors’ consideration:
1. In bar charts presented in the manuscript, it is unclear whether the error bars represent standard errors or standard deviations. The information should be indicated clearly in the figure captions. For example, see Figures 1E-1H, 2G-2H, 3D-3E, 5A-5B.
2. Figure 2:
· In Figure 2C-2D, the pollen germination tubes seem denser in sler (Figure 2D). Did the authors compare this parameter quantitatively between sler and WT?
· In Figure 2G, looking that the distribution of the data points and superimposing error bars, I am not fully convinced that there is statistically significant difference between WT and sler. I would suggest the authors to recheck this.
3. In caption of Figure 2:
· There is no mention of 2G and 2H. Please recheck.
· Is it possible to indicate more specifically the number of reps analyzed instead of just stating it is at least 10?
· For the P value, it seems redundant to indicate “0.01<P < 0.05”, instead “P< 0.05” is sufficient.
4. Figure 3 - there are no error bars in the Figure 3C unlike the rest of the bar charts in the manuscript. Please recheck. Also, did the authors perform statistical analysis on the data shown in Figure 3C-3E?
5. Figure 5B, the statistical analysis annotation (alphabetical letters above the bars) – this information could not have been obtained by simply performing two-tailed T-test. Did the authors perform any multiple comparison post hoc tests, e.g., Tukey test, to get that information? If so, it should be indicated in the caption, rather than just stating T-test was done.
6. Some statements are confusing or unclear. Please recheck. For example:
· “The height of the plant was measured and statistically based on the distance from the ground to the first flower cluster.”
· “The diameter of the stem was statistically measured …”
7. In M&M, under the section “Phenotypic measurement” - “The number of seeds per fruit and fruit weight were measured under sufficient pollination.” – Please indicate briefly what “sufficient pollination” refers to.
8. Descriptions of some of the methods used seem to have no cited references. For example, see “Aniline blue staining of pollen tubes” and “Immunolocalization of IAA”. Please recheck.
9. I would recommend including a short separate description in M&M to clearly report the statistical software and types of statistical tests used in the study.
Author Response
Responses to reviewer
- In bar charts presented in the manuscript, it is unclear whether the error bars represent standard errors or standard deviations. The information should be indicated clearly in the figure captions. For example, see Figures 1E-1H, 2G-2H, 3D-3E, 5A-5B.
The error bars in this manuscript represent standard deviations. We have added corresponding explanatory text to the figure captions. Line 180-181, 214-217, 242-243, 288-289.
- Figure 2:
- In Figure 2C-2D, the pollen germination tubes seem denser in sler(Figure 2D). Did the authors compare this parameter quantitatively between sler and WT?
Considering the observed decrease in seed number in the sler mutant, we conducted pollen germination assays to ascertain whether the reduced fertility could be attributed to abnormalities in either pollen or ovules. Our findings confirmed that normal pollen germination is not responsible for the diminished seed count in sler. After confirming normal pollen tube elongation, we did not conduct further quantitative analysis. However, considering the correlation between pollen tube elongation and auxin, this is a point of interest for future studies.
- In Figure 2G, looking that the distribution of the data points and superimposing error bars, I am not fully convinced that there is statistically significant difference between WT and sler. I would suggest the authors to recheck this.
We conducted a two-tailed t-test to identify significant differences and expanded our sample size to ensure result accuracy. The distribution of samples revealed that most fruit weights of WT were above 5g, while most of sler's fruit weights were below 4g, which ultimately leads to statistical differences. This result was reaffirmed using the statistical software OriginPro (10.1.0.170).
- In caption of Figure 2:
- There is no mention of 2G and 2H. Please recheck.
Descriptions have been added to the caption of Figure 2. Line 214-217.
- Is it possible to indicate more specifically the number of reps analyzed instead of just stating it is at least 10?
The number of repetitions for each statistic has been independently marked in the caption of Figure 2. Line 215.
- For the P value, it seems redundant to indicate “0.01<P < 0.05”, instead “P< 0.05” is sufficient.
We have corrected “0.01<P < 0.05”, to “P< 0.05”. Line 216.
- Figure 3 - there are no error bars in the Figure 3C unlike the rest of the bar charts in the manuscript. Please recheck. Also, did the authors perform statistical analysis on the data shown in Figure 3C-3E?
The data of Figure 3C was from the database tomato eFP browser (http://bar.utoronto.ca/efp_tomato/cgi-bin/efpWeb.cgi), which is a tomato transcriptome database. The original data provides RPKM values without replicates, hence no standard deviations could be calculated.
Figure 3D and E present the qPCR result of SlER we conducted in micro-Tom. Only the expression level in the roots is significantly lower than the other parts. We have added a significant difference in the revised manuscript. Figure 3E confirms that SlERL expression did not show a significant difference between WT and sler, as determined by a two-tailed t-test.
- Figure 5B, the statistical analysis annotation (alphabetical letters above the bars) – this information could not have been obtained by simply performing two-tailed T-test. Did the authors perform any multiple comparison post hoc tests, e.g., Tukey test, to get that information? If so, it should be indicated in the caption, rather than just stating T-test was done.
In Figure 5B, we used the Tukey’s test, so the t test has been corrected to the Tukey’s test. Line 289.
- Some statements are confusing or unclear. Please recheck. For example:
- “The height of the plant was measured and statistically based on the distance from the ground to the first flower cluster.”
We have deleted the “and statistically” to make the statement clear. Line 436.
- “The diameter of the stem was statistically measured…”
We have deleted the “statistically” to make the statement clear. Line 440.
- In M&M, under the section “Phenotypic measurement” - “The number of seeds per fruit and fruit weight were measured under sufficient pollination.” – Please indicate briefly what “sufficient pollination” refers to.
In this research, pollination is carried out at DAF0 via an electric vibrator TS-550 (TAKII). Then we visually confirmed enough pollen on the stigma to ensure sufficient pollination. We have added the specific pollination methods to the paper. Line 441-444. 
- Descriptions of some of the methods used seem to have no cited references. For example, see “Aniline blue staining of pollen tubes” and “Immunolocalization of IAA”. Please recheck.
“Aniline blue staining of pollen tubes” in tomato is a protocol optimized by Professor Shiba, one of the co-authors. The reference for “Immunolocalization of IAA” has been added (Line 505).
- I would recommend including a short separate description in M&M to clearly report the statistical software and types of statistical tests used in the study.
We have added it to the end of M&M part. Line 517-521.
Reviewer 2 Report
Comments and Suggestions for Authors
see PDF

moderate editing
Author Response
Responses to reviewer
Major
- Please be mindful of the language.
We will submit the final version of the manuscript to a professional English proofreader to ensure that there are no language issues.
- The concept of germination included in the Introduction is not appropriate. Likewise, in discussion you writte: germination is a critical physiological process in spermatophytes and is intricately linked to various plant hormones. It is a vague concept of germination.
We have included introductory text describing endosperm rupture in the introduction to describe the process of seed germination accurately. In the discussion, we replaced the vague description with an explanation of how auxin cooperates with ABA to inhibit germination, which is directly relevant to the core of the discussion. Line 62-63, 331-332.
- Recent updates regarding the role of auxins in seeds highlight the importance of addressing this aspect early in the Introduction, alongside recent literature on auxins and seed life. Include it!!!
We have added recent research on the impact of auxin on seed life in the introduction. Line 72-77.
- sler exhibited compact traits. What is the meaning of this title? Likewise, what does compact phenotype mean? Justify.
The miniaturization of crops has always been one of the main breeding goals and can be planted more densely to increase field utilization. The word compact is used as a suitable adjective for the trait of shrunken crops, such as the statement ”Targeting the same stem length regulator alone in groundcherry, another Solanaceae plant, also enabled engineering to a compact stature.” in the following paper (Rapid customization of Solanaceae fruit crops for urban agriculture | Nature Biotechnology). We have added an explanation of compact in the introduction. Line 139-141.
- ...pollen tubes of sler successfully germinated... (Fig. 2D). Explain this feature.
A detailed explanation has been added. Line 197-200.
- What grains are they referring to? (see Supp Fig. 2).
They are the seeds of Micro-Tom WT and sler. We have added an explanation to the caption of Supplementary figure 2.
- The legend in Fig. 4 is not sufficiently explained.
We have added an explanation to the legend in Figure 4. Line 260-266.
- Considering that auxins synthesis in seeds occurs during the early stages of endosperm development [29], we further investigated the presence of auxin in the ovary. Explain the second part of this sentence.
We have added explanatory text "auxin synthesis in seeds occurs in the endosperm immediately post-fertilization". Line 251-252.
- Additionally, the expression levels of YUC1, YUC4, YUC5, and YUC6 of sler were significantly elevated relative to those in the WT. The results of YUC1 , YUC4 in sler should be highlighted here.
According to your suggestion, we have highlighted the up-regulation of YUC1 and YUC4 expression in sler at lines 275-276.
- Fig. 5B needs to be explained more adequately.
We have added a detailed description of the expression changes of ABI3 and ABI5. Line 277-282.
- Discussions, especially in high-impact journals, must contain recent references. Additionally, it should be clear and convincing to the scientist who reads it. These characteristics reinforces the value of the discussion and justifies the implementation of the protocols developed in a draft like this. I'll give examples: [30-34, 37, 39] among other older references.
We replaced the [30,33,34,39] citations with more recent references. Here, several relatively old references were cited for specific experimental results, with no subsequent research updates available for these experiments.
- ….compromised development of megaspore mother cells. Construe it.
The megaspore mother cell is an early cell in the development of the female gamete. Normally, only one is produced in an ovule. We have added an explanation to the manuscript. Line 302-307.
- The first paragraph of the discussion is a comment on previous works, not on this draft.
In the initial paragraph of the discussion, we aimed to delve into the contrasting trajectories of functional evolution observed in the ERECTA family genes across tomato and Arabidopsis. We expected this comparison would likely engage readers’ general interest as it sheds light on the divergent evolutionary paths taken by these genes in different plant species. For the convenience of comparing the differences in gene functional divergence, detailed descriptions of previous studies in Arabidopsis were provided first.
- In pag. 7, Our investigation…. rendering it nonfunctional. I'm sorry, but I don't see enough research to make these conclusions... That is, based on the information presented in this document, I do not see enough data and fruits to support the conclusions being made. If so, could you explain it adequately?
Indeed, the [nonfunctional] is an inappropriate description. Kinases often demonstrate dosage-dependent behavior, therefore, we replaced the nonfunctional conclusion with more modest language. Line 320-325.
- The long sentence: Conversely…..thus controlling seed germination. This summarized sentence is crucial in the discussion and appears as a key conclusion in this draft. However, the authors must develop it adequately to ensure that the reader does not have interpretation issues.
We have reorganized the word order to improve the readability of the discussion. We have changed the narrative to start by introducing the inhibitory functions of auxin and ABA in seed germination and elaborate on the previous work of the AtER family in regulating seed germination through ABI3/5. Finally, we introduced the mechanism confirmed in this study by which SlER regulates auxin synthesis, affects the expression of ABI3/5, and ultimately regulates seed germination. Line 331-340.
Similar comment for this statement: We speculate that the extremely….as in tomato.
We have reorganized the word order to improve the readability of this discussion. For the statement: We speculate that the extremely….as in tomato, We have added references stating that the Arabidopsis life cycle is only 8-10 weeks. Line 353-354.
Likewise, “This contradicts the increased auxin accumulation observed in the sler mutant in our study, where we demonstrated elevated auxin levels in both the ovary and seeds (Fig. 4A and B)”. The authors attempt to justify it through previously published results (see 6, 20, 28), which does not seem appropriate without being able to compare them with non-existent results from this paper. To the authors: A discussion should be clear and precise, based obviously on the data obtained, which should be compared with previous ones, if any.
We have added narrative to the Discussion to ensure that conclusions are based on the data obtained in this study. Additionally, we have added a reference to the discussion that BR signaling in tomatoes regulates ARF5 expression and compared it with the downregulation of ARF5 expression we detected. Line 368-373.
Please keep my comment in mind. As observed, the referee has opened a parallel discussion. I request that, in the response, if deemed appropriate, the authors participate in it to address the previous discussion.
- With regard to 'ERECTA regulates fruit set' in the discussion, and to avoid repetition with what has been previously evaluated, I request that the authors clarify the role of ERECTA in tomato fruit based on results obtained in this draft (if any).
We have added fruit size data to Supplementary Figure 4. Our results show that SlER regulates tomato fruit size and weight, as well as seed number. Previous studies have not mentioned the regulation of fruit size and weight by ER. Regarding seed number, while previous studies have addressed this aspect in Arabidopsis thaliana, our hypothesis regarding the regulation of seed number by SlER through the expression of auxin response factors ARF5 and ARF7 is novel. Line 187-191.
- “Our findings elucidate the regulatory mechanism of SlER on fruit development and seed germination via auxin signaling in tomatoes” (at final of Discussion). With the results obtained in this work, this statement seems overly speculative.
We attempt to modify the statement to be not overly speculative. Line 418-420.
- Finally, the second part of the summary reflects the doubts of the referee.
Thank you for your careful suggestions. We have made revisions to the manuscript to address any concerns and minimize uncertainties.
Minor
- You consider some genes as critical genes (Fig. 5A). Why?
We have added instructions in the revised MS. The YUC genes were selected based on their rapid responsiveness to pollination. Given the limited research on the YUC family in tomatoes, we included all SlYUCs expressed in the shoots. The ARF genes was selected based on the gene expression data of the ARF family in parthenocarpy mutants in tomato. Line 75-78, 86-89, 108-109.
- That way the sentence of the Discussion is more correct: Additionally, the expression level of WT decreased in seeds soaked in water for 72 hours. However, the expression level of ABI5 in sler mutant seeds remained unchanged.
Thank you for your kind reminder. We have modified the discussion as you suggested. Line 338-339.
Reviewer 3 Report
Comments and Suggestions for Authors
The current manuscript described generation of mutation in ERECTA (SlER) gene in tomato model plant MicroTom. Authors have found that this gene play a crucail role in a numbers of processes including inflorescence development, seed number, and seed germination.
Mutant morphological and molecular characterization were performed very precisely.
However, some physiolocal investigation need to be clarify.
The title of the paper is very intriguing (seeds germination), however, the data authors provide remained many questions need to be further clarified.
Additional comments:
Line 113: punctuations.
Lines 132 – 133: authors mentioned differences in seeds germination rate between WT and mutant and this is also mention in the title. However, no protocol for seeds germination calculation were provided. Did authors do seeds priming? How they count germination? At which day? There is no also any investigation of detailed seeds and embryo structure during seeds formation, what will be very usefull for understanding the mechanism.
Figure S1 did not relate with seeds germination, but rather with seedlings growth. According to pictures of the seeds authors provided mutants have much smaller endosperm. How did author exclude that mutant seeds does not have enough carbon source to pass though soil? Hypocotyl elongation are dependent from carbon source, indeed!
Lines 172 – 180: Which auxin have you detected? Conjugated or free? Free auxin can be easy washed out during fixation (see comments to M&M). Moreover, the most importnat will be kinetics of auxin during germination process. It is still missing these.
In addition, authors have to consider that „auxin cotents“ (concentration) does not have too much biological sense since can be results of several opposite processes: high auxin may related with high synthesis+ low efllux or high synthesis+ low metabolism. Moreover, as marker of auxin signalling authrs used mainly early auxin responsive gene. These gene can not be consider as constitutive auxin responsive, indeed.
Lines 373- 382: it is better to replace four by 4. Moreover, the protocol authors used is not correct: overnight incubation in formaldehyde may lead to wash out free auxin from the samples. Authors have to used EDAC fixation (https://pubmed.ncbi.nlm.nih.gov/16546412/) in order to prevent auxin leakage.
As conclusions, I would suggest to be more accurate with description, repeat expreimnet with details seeds germination (kinetics of storage conversion to carbon, auxin deconjugation, root elongation) at the very early stages, in combination with exogenous auxin and carbon (as sucrose) applications.
Comments on the Quality of English LanguageSome puncuations.
Author Response
Responses to reviewer
Line 113: punctuations.
Lines 132 – 133: authors mentioned differences in seeds germination rate between WT and mutant and this is also mention in the title. However, no protocol for seeds germination calculation were provided. Did authors do seeds priming? How they count germination? At which day? There is no also any investigation of detailed seeds and embryo structure during seeds formation, what will be very usefull for understanding the mechanism.
Indeed, what you pointed out is an indispensable experimental condition in this paper. We have supplemented the manuscript with a germination experiment in petri dishes, providing detailed methods and results. Line 195-197, 450-457.
Figure S1 did not relate with seeds germination, but rather with seedlings growth. According to pictures of the seeds authors provided mutants have much smaller endosperm. How did author exclude that mutant seeds does not have enough carbon source to pass though soil? Hypocotyl elongation are dependent from carbon source, indeed!
We have supplemented the manuscript with a germination experiment in petri dishes (Figure 2G), which clearly demonstrates the low germination rate of sler. Considering the shrinked seed of sler (Supplementary Figure 2), we speculate that the embryos of the mutant seeds did not develop well, supporting your insight on "not having enough carbon." Here, we would like to emphasize the focus of the discussion to the hypothesis elucidated in this manuscript: that elevated auxin levels in sler seeds regulate the expression of ABI3/5, thus inhibiting seed germination through the auxin-ABA pathway. We propose to conduct carbon source-related germination experiments in future studies, perhaps this could further focus the readers' attention on the topic of "the regulation of seeds and fruits by ERECTA through auxin signaling in tomato". Line 356-359.
Lines 172 – 180: Which auxin have you detected? Conjugated or free? Free auxin can be easy washed out during fixation (see comments to M&M). Moreover, the most importnat will be kinetics of auxin during germination process. It is still missing these.
The auxin detected in the seed endosperm in our study is predominantly in the conjugated form. Previous studies have demonstrated that only a minimal portion of indole-3-acetic acid (IAA) is present in its free form in mature seeds. For instance, in mature bean seeds, free auxin constitutes less than 6% (source: https://academic.oup.com/plphys/article/91/2/775/6085128). Therefore, we suggest that most of the auxin synthesized in the endosperm remains in the seed and inhibits seed germination through the ABA pathway.
In addition, authors have to consider that „auxin cotents“ (concentration) does not have too much biological sense since can be results of several opposite processes: high auxin may related with high synthesis+ low efllux or high synthesis+ low metabolism. Moreover, as marker of auxin signalling authrs used mainly early auxin responsive gene. These gene can not be consider as constitutive auxin responsive, indeed.
Previous studies have concluded that the auxin biosynthesis level in erecta mutants decreased based on the intensity of the DR5::GFP signal. The upregulation of YUCs expression in the ovary and the higher auxin levels than WT in sler seeds detected by immunolocalization in this manuscript can verify from initiation to fruition, that the upregulation of auxin synthesis in the ovary stage in sler ultimately leads to excess auxin accumulation. This study provides reliable results through multiple validations (immunolocalization and the expression levels of auxin synthesis genes), which can critically revise previous research conclusions.
Lines 373- 382: it is better to replace four by 4. Moreover, the protocol authors used is not correct: overnight incubation in formaldehyde may lead to wash out free auxin from the samples. Authors have to used EDAC fixation (https://pubmed.ncbi.nlm.nih.gov/16546412/) in order to prevent auxin leakage.
We have made corrections to the English writing in the revised manuscript, such as replacing four by 4. Line 501-502.
Thank you for your correction. We will pay attention to the use of fixatives in future studies and their impact on the results. The detected auxin signal intensity in this study can reflect the conjugated auxin accumulation in the seed endosperm. Experiments on both WT and sler mutants were conducted under strict identical conditions. Therefore, we believe the obtained results can objectively reflect the relative quantification of auxin accumulation.
Reviewer 4 Report
Comments and Suggestions for Authors
Dear authors,
The manuscript "ERECTA Modulates Seed Germination and Fruit Development via Auxin Signaling in Tomato" brings innovative and important information regarding the gene ERECTA for tomato breeders. In general, the manuscript is well presented and the results support the main hypothesis.
Review notes:
1 - Introduction - Provide citation to the sentence "Excessive seed content in tomatoes can negatively impact the taste"
2 - Results - sler exhibited compact traits - Write full name to the abbreviation DAF0 as shown for the first time in the text.
3 - Result - sler altered fruit and seed formation - Remove "than" in the sentence "reduction compared to than that of the WT"
- Correct "t" in the sentence "whereas t sler mutant fruits contained 7.22±4.86"
4 - Results - The subsection "Genetic redundancy between SlER and AtER" could be the first subtopic.
Figure 3 - Increase font size in the Y-axis labels.
5 - Figure 5B - Put WT and sler on the same Y-axis scale.
6 - Discussion - ERECTA regulates seed germination through auxin signaling - Correct "acritical" in the sentence "germination is acritical physiological process"
- add commas to the sentence "the ater aterl1 and aterl2 triple mutant"
7 - Material and methods - RNA extraction and qRT-PCR - add how the relative expression values were obtained.
Author Response
Responses to reviewer
1 - Introduction - Provide citation to the sentence "Excessive seed content in tomatoes can negatively impact the taste"
We have added the citation to the sentence. Line 112.
2 - Results - sler exhibited compact traits - Write full name to the abbreviation DAF0 as shown for the first time in the text.
The full name of DAF days after flowering has been added. Line 168.
3 - Result - sler altered fruit and seed formation - Remove "than" in the sentence "reduction compared to than that of the WT"
The "than" has been removed from the sentence. Line 185.
- Correct "t" in the sentence "whereas t sler mutant fruits contained 7.22±4.86"
It has been corrected. Line 193.
4 - Results - The subsection "Genetic redundancy between SlER and AtER" could be the first subtopic.
Thank you for your suggestions regarding the order of the results in the manuscript. We propose to include a detailed description of the phenotype in the initial section to ensure that readers have a comprehensive understanding of the characteristics of the sler mutant. Additionally, the first part of the discussion will integrate the phenotypic data with fertilization and fruit formation. Therefore, we hope to maintain the current order of description.
Figure 3 - Increase font size in the Y-axis labels.
We unified the font size of the Y-axis, and increased font size of X-axis.
5 - Figure 5B - Put WT and sler on the same Y-axis scale.
We have unified the scale of the Y-axis.
6 - Discussion - ERECTA regulates seed germination through auxin signaling - Correct "acritical" in the sentence "germination is acritical physiological process"
It has been corrected to “a critical”. Line 331.
- add commas to the sentence "the ater aterl1 and aterl2 triple mutant"
We have corrected "the ater aterl1 and aterl2 triple mutant" to “the ater aterl1 aterl2 triple mutant” following the writing style specified in the paper (https://academic.oup.com/plcell/article/34/10/3665/6650632). Line 349.
7 - Material and methods - RNA extraction and qRT-PCR - add how the relative expression values were obtained.
The calculation method of relative expression and the internal reference genes used have been supplemented in the text. Line 513-515.
Round 2
Reviewer 2 Report
Comments and Suggestions for Authors
The Referee objections have been adequately adressed. Therefore, I give my consent to proceed with publishing this draft.
Author Response
We sincerely appreciate your valuable suggestions.
Reviewer 3 Report
Comments and Suggestions for Authors
Thank you for response. The text is better and can be provisionaly accepted, after some corrections .
See point below.
My best regards.
Line 170: after (DAF0) should be „.“ And new sentence.
Lines 276 – 277: did you mean SlYUC gene?
Please, consider that each YUC (FZY) gene have hos own localization and bfunction and expressed during growth as cascade dependent from the process currently predominant. Please, look here: Expósito-Rodríguez, M., Borges, A. A., Borges-Pérez, A., Hernández, M., & Pérez, J. A. (2007). Cloning and biochemical characterization of ToFZY, a tomato gene encoding a flavin monooxygenase involved in a tryptophan-dependent auxin biosynthesis pathway. Journal of plant growth regulation, 26, 329-340.
Similar with ARF/IAA gene. That’s why the usefull information can be obtained only after investigation of several time point. Please, keep it in mind for the future.
Line 361: „In addition to the inhibition of germination by the auxin ABA pathway, the seed size of sler is smaller than that of WT“ ??? Why in addition? Seeds smaller is a fact and must be mention firstly.
Line 334: auxin has been shown to enhance the inhibition of germination“ ??? This suggestion is not true. Seeds germination (radicle growth) is re-establishmnent of auxin transport and canalization and induced cell elongation in the embronic root tips. Maybe authors mean inhibition of auxin transport by ABA or inhibition of auxin response? Both processes is an opposite to auxin accumulation.
It will be great to check role of erecta in seeds germination to study re-establishmnent of auxin transport in embryonic root after seeds germination, indeed. But it is future task, of course.
Maybe authors can include this point in discussion.
Line 462: seven days is too long. Germination should be counted after 3. 5 and 7 days in order to see kinetics. I also would suggest fort he next test make a very liitle mechanical damage to have a better germination (without damaging embryo, of course).
Line 487: „sequence of SlER was amplified“ ? Sequence can not be amplify itself. cDNA can be.
Comments on the Quality of English LanguageMinor punctuation corrections.
Author Response
Responses to reviewer
Line 170: after (DAF0) should be „.“ And new sentence.
We have corrected it to “at 0 days after flowering (DAF0). The”. Line 169.
Lines 276 – 277: did you mean SlYUC gene?
Please, consider that each YUC (FZY) gene have hos own localization and bfunction and expressed during growth as cascade dependent from the process currently predominant. Please, look here: Expósito-Rodríguez, M., Borges, A. A., Borges-Pérez, A., Hernández, M., & Pérez, J. A. (2007). Cloning and biochemical characterization of ToFZY, a tomato gene encoding a flavin monooxygenase involved in a tryptophan-dependent auxin biosynthesis pathway. Journal of plant growth regulation, 26, 329-340.
A correspondence table of YUC and FZY is added to avoid ambiguity in Supplementary Table 2.
Referring to this study on the expression of FZY, FZY2 (YUC4), FZY3 (YUC7), and FZY6 (YUC6) have higher expression levels during the flowering stage (https://www.sciencedirect.com/science/article/pii/S0981942811000787?via%3Dihub). In our study, FZY2 (YUC4) and FZY6 (YUC6) were detected to have up-regulated expression in the sler ovary. The earlier version of the manuscript did not incorporate the expression data of FZY3 (YUC7) as its expression was not detected in the ovary, and we have added the data of FZY3 (YUC7) to Fig. 5A. The modifications regarding this section of the manuscript are reflected in Line 89-91 and 350-356.
Similar with ARF/IAA gene. That’s why the usefull information can be obtained only after investigation of several time point. Please, keep it in mind for the future.
Thank you for your kind reminder.
Line 361: „In addition to the inhibition of germination by the auxin ABA pathway, the seed size of sler is smaller than that of WT“ ??? Why in addition? Seeds smaller is a fact and must be mention firstly.
In the revised manuscript, the content regarding seed size has been relocated to an earlier position in the discussion section to emphasize its significance. Lines 334-339.
Line 334: auxin has been shown to enhance the inhibition of germination“ ??? This suggestion is not true. Seeds germination (radicle growth) is re-establishmnent of auxin transport and canalization and induced cell elongation in the embronic root tips. Maybe authors mean inhibition of auxin transport by ABA or inhibition of auxin response? Both processes is an opposite to auxin accumulation.
Several previously regarded as reliable studies have indicated that excess auxin stored in seeds can inhibit germination via ABA (https://www.pnas.org/doi/full/10.1073/pnas.1304651110 and the subsequent study of the previous paper https://academic.oup.com/plcell/article/35/3/1110/6896260). In the discussion, the viewpoint regarding auxin enhance the inhibition of germination is derived from these previous studies.
It will be great to check role of erecta in seeds germination to study re-establishmnent of auxin transport in embryonic root after seeds germination, indeed. But it is future task, of course.
Maybe authors can include this point in discussion.
We have added it to the discussion. Line 364-366.
Line 462: seven days is too long. Germination should be counted after 3. 5 and 7 days in order to see kinetics. I also would suggest fort he next test make a very liitle mechanical damage to have a better germination (without damaging embryo, of course).
Thank you for your valuable suggestions. That will be an interesting experment in future research.
Line 487: „sequence of SlER was amplified“ ? Sequence can not be amplify itself. cDNA can be.
We have changed the sentence to active to avoid ambiguity. Line 486-488.